# The Impact of Prebiotic, Probiotic, and Synbiotic Supplements and Yogurt Consumption on the Risk of Colorectal Neoplasia among Adults: A Systematic Review

**DOI:** 10.3390/nu14224937

**Published:** 2022-11-21

**Authors:** Claire E. Kim, Lara S. Yoon, Karin B. Michels, Wynn Tranfield, Jonathan P. Jacobs, Folasade P. May

**Affiliations:** 1Department of Epidemiology, Harvard T.H. Chan School of Public Health, Boston, MA 02115, USA; 2Department of Epidemiology, Fielding School of Public Health, University of California, Los Angeles, CA 90095, USA; 3Institute for Prevention and Cancer Epidemiology, Faculty of Medicine and Medical Center, University of Freiburg, 79098 Freiburg, Germany; 4University Library, University of California, Santa Cruz, CA 95064, USA; 5Department of Medicine, David Geffen School of Medicine, University of California, Los Angeles, CA 90095, USA; 6The Vatche and Tamar Manoukian Division of Digestive Diseases, Department of Medicine, David Geffen School of Medicine, University of California, Los Angeles, CA 90095, USA; 7Division of Gastroenterology, Department of Medicine, VA Greater Los Angeles Healthcare System, Los Angeles, CA 90073, USA

**Keywords:** prebiotic, probiotic, yogurt, gut microbiome, colorectal neoplasia

## Abstract

Prebiotic and probiotic supplementation and yogurt consumption (a probiotic food) alter gut microbial diversity, which may influence colorectal carcinogenesis. This systematic review evaluates the existing literature on the effect of these nutritional supplements and yogurt consumption on colorectal neoplasia incidence among adults. We systematically identified ten randomized controlled trials and observational studies in adults age ≥ 18 without baseline gastrointestinal disease. Prebiotics included inulin, fructooligosaccharides, galactooligosaccharides, xylooligosaccharides, isomaltooligosaccharides, and β-glucans. Probiotics included bacterial strains of Lactobacillus, Bifidobacterium, Saccharomyces, Streptococcus, Enterococcus, Bacillus, Pediococcus, Leuconostoc, and Escherichia coli. Synbiotic supplements, a mixture of both prebiotic and probiotic supplements, and yogurt, a commonly consumed dietary source of live microbes, were also included. We defined colorectal neoplasia as colorectal adenomas, sessile serrated polyps, and colorectal cancer (CRC). Overall, findings suggest a moderate decrease in risk of adenoma and CRC for high levels of yogurt consumption compared to low or no consumption. Prebiotic supplementation was not associated with colorectal neoplasia risk. There was some evidence that probiotic supplementation may be associated with lower risk of adenomas but not with CRC incidence. Higher yogurt consumption may be associated with lower incidence of colorectal neoplasia. We found little evidence to suggest that prebiotic or probiotic supplements are associated with significant decreases in CRC occurrence.

## 1. Introduction

Colorectal cancer (CRC) is the third most commonly diagnosed cancer and second most common cause of cancer-related deaths among men and women in the United States (US) [1]. Advances in screening have contributed to declining incidence rates over time through early detection and subsequent removal of colorectal adenomas and other precursors to CRC. Lifestyle modification, including efforts to reduce obesity, tobacco use, and intake of processed and red meats, are also emphasized to reduce CRC risk and recurrence [2,3].

Clinical and epidemiologic studies suggest a chemopreventive effect of prebiotic, probiotic and synbiotic supplements on colorectal neoplasia (e.g., precancerous polyps, colorectal adenoma and CRC). Prebiotics are nondigestible food ingredients that improve host health by selectively stimulating the growth and activity of bacteria in the colon, collectively known as the gut microbiome [4]. Prebiotics’ role in modulating lipid metabolism, producing short chain fatty acids, balancing intestinal pH, lymphocytes and leukocytes in the lymphoid tissues, enhancing nutrient absorption, and shortening fecal transit time may reduce exposure to carcinogens and tumor promoters [4,5,6,7]. Probiotics are live bacteria and yeasts found in various functional foods and supplements that help promote beneficial bacteria in the digestive tract once consumed [8]. They have been used to facilitate the re-establishment of the gut microbiome after disruption by antibiotics or infection, and to enhance diversity of the gut microbial communities [9]. Yogurt, which traditionally contains two species of lactic acid bacteria, *Lactobacillus delbrueckii* ssp. *Bulgaricus* and *Streptococcus thermophilus*, is thought to have probiotic properties [10].Yogurt is easily accessible and popularly consumed for their potential health benefits to the gut microbiome [11]. Synbiotic supplements are a mixture of prebiotic and probiotic supplements taken with the goal to achieve the health benefits of both simultaneously or synergistically [12].

Previous systematic reviews have examined the impact of prebiotics and probiotics on gut microbial or CRC outcomes, including colonic inflammatory markers, microbial diversity, and postoperative complications after CRC resection [13,14,15,16]. However, no reviews to date have focused on the effect of prebiotic and probiotic supplementation on incidence of colorectal neoplasia in population-based studies. Our main objective was to systematically evaluate the existing literature on the effect of prebiotic, probiotic and synbiotic supplements (including yogurt) on the incidence of colorectal neoplasia among adults.

## 2. Materials and Methods

The eligibility criteria for this systematic review were informed by the PICOS (population, intervention, comparison, outcome and study design) guidelines [17]. We conducted a comprehensive literature search in July 2021 to identify studies in English published between January 1966 and July 2021. Search strategies were designed in collaboration with a librarian. Searches included controlled vocabulary and text word terms describing probiotic, prebiotic, synbiotic supplements and yogurt consumption influence on colorectal neoplasia and associated conditions (Appendix A). The search included bibliographic databases (Embase, Cochrane Reviews and Trials, PubMed) and grey literature (ClinicalTrials.gov, Google Scholar). Reference lists of all included studies were screened to identify additional relevant studies. The current review is registered in PROSPERO (CRD42020196584).

The primary outcome of interest was incident colorectal neoplasia, including adenomas, sessile serrated polyps, and CRC. The target study population was adults age ≥ 18 without baseline gastrointestinal disease. Studies were excluded if the study population included adults diagnosed with inflammatory bowel disease (Crohn’s disease, ulcerative colitis), irritable bowel syndrome, and other gastrointestinal disorders. We included studies with adults without a history of CRC or a history of CRC resection within 3 months of the respective studies.

We included four exposure types: prebiotic supplementation, probiotic supplementation, synbiotic supplementation, and yogurt consumption. Prebiotic supplements included inulin, fructooligosaccharides (FOS), galactooligosaccharides (GtOS), xylooligosaccharides (XOS), isomaltooligosaccharides (IMO), and *β*-glucans. Probiotic supplements included bacterial strains of the following genera: *Lactobacillus*, *Bifidobacterium*, *Saccharomyces*, *Streptococcus*, *Enterococcus*, *Bacillus*, *Pediococcus*, *Leuconostoc*, and *Escherichia coli.* Synbiotic supplements consist of both prebiotic and probiotic components. Yogurt is a popularly consumed dietary source of live microbes, namely lactic acid-producing bacteria, and was therefore included in our search. We excluded studies that grouped yogurt consumption with other dairy products (e.g., milk, cheese). We also excluded interventions that included antibiotics and other medications.

Search filters included randomized controlled trials and observational studies (cohort, case–control, cross-sectional, or case series). We included studies in which the comparator(s) or control group was a “regular” diet or placebo.

Two reviewers (CK and LY) worked independently to evaluate studies for inclusion. First, titles and abstracts were assessed for inclusion based on eligibility criteria (Figure 1). Eligible manuscripts proceeded to the full text screening. Full text manuscripts that met inclusion criteria were chosen for data extraction. A third reviewer (FM) was consulted in cases of disagreement between the two primary reviewers on eligibility. We used Covidence, a systematic review management software, to facilitate this process [18].

The same two reviewers independently performed data extraction for selected manuscripts. Reviewers used a standardized data collection form to extract journal name, year, research question, hypotheses, study design, recruitment method, inclusion and exclusion criteria for participants, intervention characteristics (e.g., type, duration), outcome measures, statistical methods, results, and methodological limitations. All data were recorded into a single excel spreadsheet.

For quality assessment, we adapted the Cochrane Risk of Bias 2 (RoB 2) tool for randomized studies and Risk of Bias in Non-Randomized Studies of Interventions (ROBINS-I) criteria from Sterne et al. for non-randomized studies [19,20]. We assessed blinding methods (double, single, or none) and potential for information bias via exposure (prebiotic, probiotic, and synbiotic supplements) measurement and outcome ascertainment for relevant studies. We assessed for selection bias by checking the number of patients lost to follow-up over the study periods. The two reviewers completed all risk of bias assessments independently.

From clinical trials and cohort studies, we extracted the relative risk (RR) or hazard ratio (HR) of colorectal neoplasia, comparing prebiotic, probiotic, and synbiotic supplement groups to placebo or other comparison groups. From case–control and cross-sectional studies, we extracted odds ratios (OR).

## 3. Results

A summary of the literature search and inclusion criteria is provided in Figure 1. Overall, 3364 articles were identified in PubMed, Embase, Web of Science, Cochrane Reviews, and Clinical Trials. Of these, 2542 were excluded based on the title or abstract, an additional 70 were excluded after full text review, and 742 duplicates were excluded. The primary reasons for exclusion were ineligible study design (e.g., animal models, cell-line analyses), study populations (e.g., patients undergoing CRC surgery), exposure or intervention, and outcome. Several studies evaluated suspected biomarkers for CRC risk, such as gut microbial composition, anti-inflammatory response, or proliferation of colorectal cells. These putative markers for CRC were not included in this review.

Table 1 presents a summary of study characteristics for the ten studies included in the systematic review. Among the ten studies included, five were conducted in Europe, four in the US, and one in Japan. Participants across all studies were at least 30 years old at the time of enrollment, and study sample size ranged from 398 to 160,195. Three studies evaluated prebiotic supplementation, one evaluated probiotic supplementation, six evaluated yogurt consumption, and one evaluated both probiotic supplements and yogurt consumption in relation to CRC incidence.

### 3.1. Studies on Prebiotics and Probiotics

Two prospective cohort studies and a randomized controlled study evaluated the role of prebiotic fiber on CRC risk. Castro-Espin et al. conducted a sub-analysis of 53,700 adult men and women participating in the European Prospective Investigation into Cancer and Nutrition (EPIC) Oxford cohort study [21]. Study participants reported frequency of prebiotic fiber intake (total prebiotic fiber, fructan, and galacto-oligosaccharide [GOS]) using semi-quantitative food frequency questionnaires and were followed for 16 years to determine CRC incidence. A total of 574 CRC cases were identified. The highest quartile of total prebiotic fiber, fructan, and GOS (compared to the lowest quartile) intake was not associated with risk of CRC. The study was published in abstract form at the Proceedings of the Nutrition Society European Nutrition Conference in 2020; additional details on prebiotic exposure and data stratified by sex were not available at the time of review.

The second prospective cohort study assessed prebiotic fiber intake on incident CRC among 160,195 women in the US-based Women’s Health Initiative [22]. Approximately 3.7% of the cohort was a user of any prebiotic fiber supplement (total prebiotic fiber, soluble prebiotic fiber, insoluble prebiotic fiber) (*n* = 5944). Most prebiotic supplement users reported using soluble psyllium fiber. The study followed the women for 15 years and assessed incident CRC using self-report and medical chart review and CRC mortality using medical record and death certificate review. They found that any prebiotic supplement use at baseline was not associated with risk of CRC; however, use of insoluble fiber was associated with higher CRC-related mortality compared to non-use.

One randomized controlled study evaluated both prebiotic and probiotic supplementation. Ishikawa et al. conducted a randomized controlled trial to evaluate the effect of prebiotic fiber and Lactobacillus casei administration on the occurrence of colorectal tumors (adenomas and/or early cancers) among Japanese adults with a history of CRC tumors [23]. A total of 398 study participants free from other tumors at baseline were randomized into one of four intervention groups: prebiotic fiber (dietary wheat bran) (25 g per day), *L. casei* (1 g per meal), both or neither. At the end of the four-year study, participants were evaluated for incident colorectal tumors. The study did not report a significant difference in occurrence of tumors for those randomized to the *L. casei* group nor the wheat bran group when compared to the no treatment group.

A case–control study from Rifkin et al. compared users and non-users of probiotic supplementation on incident and recurrent colorectal polyps in a population of adult men and women participating in the Tennessee Colorectal Polyp Study (TCPS) and the Johns Hopkins Biofilm Study (JHBS) [24]. Among those participating in the JHBS, weekly probiotic supplementation (1–6 days/week) was associated with lower odds of colorectal polyps compared to no probiotic use in the past 12 months.

### 3.2. Studies on Yogurt

Seven studies evaluated the association between yogurt consumption and colorectal neoplasia. Rifkin et al., also evaluated yogurt consumption in both the TCPS and the JHBS. In the TCPS, the authors found that daily (average 0.1 cup per day) compared to no or rare intake was associated with lower odds of hyperplastic polyps [24]. Furthermore, weekly yogurt intake (1 cup serving size consumed more than once a week), compared to no or rare intake, was associated with lower odds of adenomatous polyps among adults with a history of colorectal neoplasia in TCPS. In the JHBS, both weekly yogurt intake and probiotic use were not associated with colorectal polyps among healthy adults [24]. One other case–control study evaluated the association of yogurt with colorectal neoplasia: Senesse et al. conducted a study among adults in France to assess yogurt consumption in relation to small and large adenomatous polyps [28]. The study found that high yogurt (greater than 68.9 g/2 wks in men; greater than 71.2 g/2 wks in women) consumption one year prior to diagnosis was associated with lower risk of large adenomas but not associated with small adenoma occurrence.

The remaining five yogurt studies were prospective cohort studies in the US and Europe. A study by Zheng et al. evaluated yogurt consumption and conventional adenomas and serrated lesions among men participating in the US-based Health Professionals Follow-up Study. In this study, higher baseline yogurt consumption (>2 cup servings per week) compared to no consumption was associated with lower risk of conventional adenomas but not serrated lesions over a 24-year period among men. There was no association of yogurt consumption with conventional adenomas or serrated lesions among women. Another study assessed yogurt consumption with CRC incidence and mortality among women participating in the Nurses’ Health Study [30]. Michels et al. found that baseline yogurt consumption of 1+ cup serving/week, compared to never or <1 cup serving/month, was associated with lower risk of CRC but not CRC-related mortality over a 32 year period [30]. Similarly, among participants in EPIC cohorts of study samples of 477,122 participants and 45,241 participants, higher yogurt consumption (>17.8 g/day for one study and 71 g/day for the other) was associated with lower relative risk of incident CRC compared to no yogurt consumption [27,29]. Finally, Barrubes et al. conducted a study of 7447 participants at high risk of cardiovascular disease to evaluate dairy product consumption, including yogurt and fermented dairy, and CRC risk [25]. In the 6-year follow-up period, 101 incident CRC cases were identified, however yogurt (8 g vs. 65 g) and fermented dairy consumption (206 g vs. 350 g) were not associated with CRC incidence.

### 3.3. Studies on Synbiotics

No studies evaluated the association between synbiotic supplementation and risk of colorectal neoplasia.

## 4. Discussion

The purpose of this systematic review was to characterize the association of prebiotics, probiotics, synbiotic supplements and yogurt consumption with incidence of colorectal neoplasia (adenoma, sessile serrated polyps, CRC). Previous reviews of prebiotics, probiotics, synbiotics, or yogurt have examined their effects on putative CRC markers or outcomes related mechanistically to CRC, such as gut microbial composition, immune system modulation, cell proliferation, or post-surgical quality outcomes in preclinical studies, in observational studies, and clinical trials; yet, none have focused on CRC neoplasia incidence [13,14,16,31]. The cumulative findings from human population-based studies included in this systematic review suggest that higher yogurt consumption may be associated with lower incidence of colorectal neoplasia. Overall, prebiotic supplementation was not associated with risk of colorectal neoplasia. The findings also suggest that probiotic supplementation may be associated with lower risk of adenomas but not with CRC incidence.

These results do not support findings from prior studies of pre-clinical models and human biomarker studies that prebiotic and probiotic supplementation may help prevent CRC [13,14,16]. There are several differences in the studies included in our review and those included in previous reviews. Notably, previous reviews included studies that measured indirect markers of CRC risk, such as composition and functional activity of gut microbiota, immune activity, or quantification of crypt foci, rather than incident adenoma or CRC. These studies provide mechanistic evidence to suggest a protective effect of prebiotics and probiotics on CRC but do not evaluate the distal outcome of colorectal neoplasia. In addition, many of the studies included in other reviews were performed in animal models or used cell-lines to evaluate biological responses related to CRC. A systematic review of the mechanistic role of probiotic and synbiotic supplements in colorectal carcinogenesis identified a total of 33 studies published through 2018 [14]. Only 3 of the included studies were done in humans; of those 3, only 1 evaluated incident CRC. The limited population-based evidence evaluating the effect of dietary supplementation on incident CRC is not surprising given the high costs, large sample sizes, and long follow-up periods required for prospective or interventional studies.

Prior studies suggest that prebiotic fiber is associated with putative biomarkers of CRC and microbiome-related outcomes. A study by Roncucci et al. assessed the effect of daily consumption of lactulose on adenoma recurrence in adults and reported significant reduction in adenoma recurrence among those with lactulose supplementation [32]. Clinical studies have also reported beneficial effects of resistant starch on CRC risk through reduction in the percent of mitotic cells in the crypt as well as overexpression of genes, CDK4 and GADD45A, which are associated with reduction in cell proliferation and increased genomic stability [33]. Additionally, a systematic review by Clark et al., summarized nine randomized controlled trials evaluating the effect of prebiotics on biomarkers of CRC in humans. Three of the nine studies reported beneficial effect of prebiotics including a reduction in adenoma recurrence, favorable crypt mitotic location and gene expression, and a minor increase in DNA methylation [13].

Colon tumorigenesis has been associated with gut microbiome dysbiosis and increased intestinal permeability, favoring bacterial translocation through the mucosal epithelium and production of proinflammatory cytokines [34]. These findings suggest an opportunity to counteract tumorigenesis by promoting a healthy balance of the gut microbiome by administering prebiotics and/or selective beneficial bacteria (probiotics). Specifically, animal studies have reported that bacterial metabolites of prebiotic fermentation and of Lactobacillus and Bifidobacterium probiotics, including butyrate, acetate and propionate, may modulate inflammation, epithelial barrier repair and proliferation, and tumor cell apoptosis [35,36]. However, these results are difficult to evaluate given the use of different prebiotic fibers and probiotic strains across studies and variability in their administration, including dose, frequency, timing, and delivery method (e.g., via supplements, yogurt) [34]. Moreover, given the large individual variability in the microbiome, a ‘standard’ administration of probiotics may not benefit each individual, further complicating our understanding of the role of probiotics [37]. It is also important to consider the current challenges in optimizing the full beneficial effect of probiotic supplements in regard to appropriate strain selection, process and storage conditions, cell viability and functionality, and effective delivery at the target site [38]. More human studies are needed to substantiate these mechanisms as potential pathways for CRC prevention.

Yogurt, a food source containing live microbes, has been suggested to influence CRC risk through microbiome, immune, or inflammatory pathways [24,26,30]. Specifically, lactic acid bacteria, which expand during the fermentation process of yogurt, have been reported to have antihypertensive, antimicrobial, antioxidative, and immune-modulatory properties [39]. Yogurt is also high in calcium which may modulate oxidative DNA damage, induce apoptosis, hinder proliferation of colonic epithelium cells, and inhibit heme-induced colon carcinogenesis [40,41,42,43]. Additionally, vitamin D in yogurt may not only inhibit cell proliferation but also promote greater calcium absorption [44,45,46]. Other bioactive components synthesized include butyric acid and conjugated linoleic acid (CLA). In vitro studies reported butyric acid to potentially induce differentiation in a wide range of tumor cell lines [47,48,49] while CLA decreased the number of aberrant crypt foci in rats that were under 2-amino-3-methylimidazo [4,5-f] quinoline-induced colon carcinogenesis [50]. Few experimental studies have evaluated yogurt as a probiotic exposure due to feasibility and possible residual confounding by other nutritional components (e.g., B vitamin folate, riboflavin, and B12) in yogurt [51].

While the current systematic review is a comprehensive summary of the current literature on the association of prebiotic, probiotic, synbiotic supplements and yogurt consumption with CRC risk, the included studies varied in population, study design, and analysis methods. The methodological variability of the included studies did not allow for meta-analysis of study results and made it difficult to characterize each exposure intake with CRC risk. Our research question, with specific exposure and outcome interests, limited the number of available studies to be included. Specifically, there were very few studies with an incidence of colorectal neoplasia outcome as such would require a long follow-up time. It is also important to note that our study focused on prebiotic and probiotic supplements and yogurt and did not include other prebiotic and probiotic-related foods such as fermented vegetables. Lastly, we cannot overlook the potential for publication bias, favoring potential significant findings over null findings.

Nonetheless, there are many strengths to this study. It is the first population-based systematic review to comprehensively assess the association of prebiotics, probiotics, probiotic, synbiotic supplements, and yogurt consumption with colorectal neoplasia incidence. We have conducted a highly thorough search process and selected literature based on widely recommended and approved practices. We also ensured the quality of the studies included through the employment of Cochrane risk of bias analyses. The work addresses an important question in CRC prevention and informs clinical recommendations.

## 5. Conclusions

Our findings suggest a possible association between higher yogurt consumption and lower incidence of colorectal neoplasia. Prebiotic supplementation was not associated with colorectal neoplasia risk. There was some evidence that probiotic supplementation may be associated with lower risk of adenomas but not with CRC incidence. Further research is needed to understand whether the association between yogurt and colorectal neoplasia is due to the microbial content of yogurt or other residual nutritional components. Understanding the mechanistic pathways that explain the potential casual effect between yogurt and colorectal neoplasia may inform CRC preventive efforts and research towards understanding the etiology for other types of neoplasia.

## Figures and Tables

**Figure 1 nutrients-14-04937-f001:**
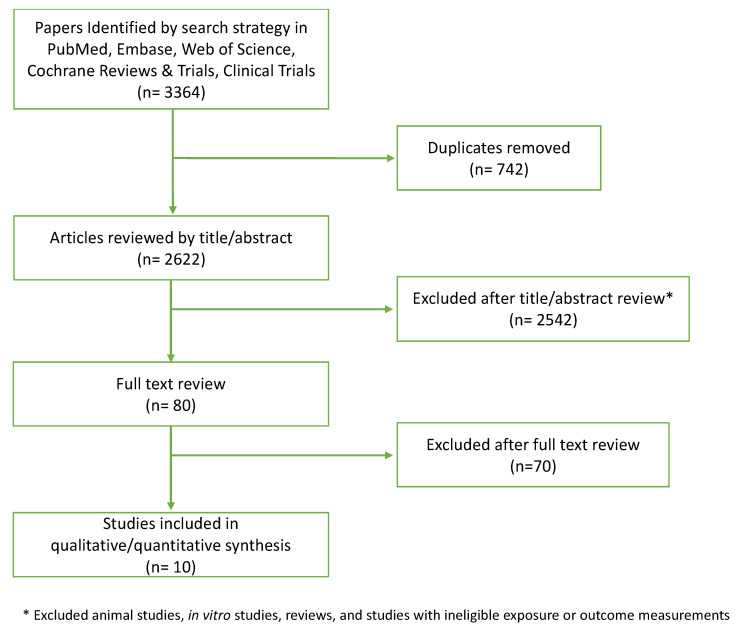
PRISMA diagram depicting screening process.

**Table 1 nutrients-14-04937-t001:** Description of studies included in the current systematic review.

References	Study Population	Study Design	Sample Size	Prebiotic, Probiotic, Yogurt	Dose and Duration	Follow-Up Period (years)	Primary Outcome	Comparison	Effect Size	Key Result
Castro-Espin et al. [21]	Adult men and women aged 35–69 years living in England and Scotland (EPIC-Oxford study)	Prospective cohort	53,700	Prebiotic fiber (total, fructan, GOS)	Frequency of intake at baseline (daily)	16.1	Incident CRC	Total prebiotic: Highest intake quartile compared to lowest intake quartile	HR 0.87; 95% CI 0.66–1.14	Total prebiotic intake (total, fructan-specific, and GOS-specific) was not associated with CRC risk.
								Total fructan: Highest intake quartile compared to lowest intake quartile	HR 0.91; 95% CI 0.70–1.19	
								Total GOS: Highest intake quartile compared to lowest intake quartile	HR 0.87; 95% CI 0.66–1.15	
Skiba et al. [22]	Adult women aged 50–79 participating in the Women’s Health Initiative in the United States without a history of CRC	Prospective cohort	160,195	Prebiotic fiber (total, soluble, insoluble)	User versus nonuser at baseline	15.4	Incident CRC	Total prebiotic: User (any amount) compared to non-user	HR 1.12; 95% CI 0.91–1.38	Total prebiotic supplement use was not associated with CRC risk or mortality, but insoluble fiber was associated with higher CRC mortality.
								Soluble prebiotic: User (any amount) compared to non-user	HR 1.08; 95% CI 0.87–1.34	
								Insoluble prebiotic: User (any amount) compared to non-user	HR 1.48; 95% CI 0.87–2.51	
Ishikawa et al. [23]	Adult men and women aged 40–65 years with 2+ CRC tumors removed in Japan	Randomized clinical trial	398	Probiotic (*L. casei)*	1 g per meal	4	Incident CR tumors	*L. casei* group compared to no treatment *group*	OR 0.85; 95% CI 0.56–1.27	There was no significant difference in risk of new CR tumors for those randomized to the *L. casei* group nor the wheat bran group when compared to the no treatment group.
				Prebiotic fiber (Wheat bran biscuits)	25 g per day			Wheat bran group compared to no treatment group	OR 1.31; 95% CI 0.87–1.97	
Rifkin et al. [24]	Adult men and women aged 40–75 (TCPS) and 40–85 (JHBS) undergoing routine colonoscopy without a history of CRC or inflammatory bowel disease	Case-control	5446 (TCPS) 1061 (JHBS)	Yogurt	Frequency of intake (daily, weekly, monthly, none)	-	Incident or recurrent colorectal polyps	Yogurt: Daily intake compared to no/rare intake (TCPS)	OR 0.54; 95% CI 0.31–0.95	Daily yogurt intake was associated with lower odds of colorectal polyps.
								Yogurt: Weekly use compared to no use (JHBS)	OR 0.75; 95% CI 0.54–1.04	
				Probiotic supplementation	Use versus non-use in the past week			Probiotic: Weekly use compared to no use (JHBS)	OR 0.72; 95% CI 0.49–1.06	
Barrubes et al. [25]	Adult men and women aged 55–80 at high risk of CVD participating in the PREDIMED study	Prospective cohort	7447	Yogurt	Frequency of intake (g per day)	6	Incident CRC	Yogurt: Highest intake tertile compared to lowest intake tertile	HR 0.94; 95% CI 0.56–1.59	Yogurt and fermented dairy consumption were not associated with CRC.
				Fermented dairy				Fermented dairy: Highest intake tertile compared to lowest intake tertile	HR 0.90; 95% CI 0.53–1.53	
Zheng et al. [26]	Adult men participating in the Health Professionals Follow-up Study (HPFS) and women participating in the Nurses’ Health Study (NHS) aged 35 + years	Prospective cohort	32,606 (men) 55,743 (women)	Yogurt	Frequency of intake (servings per week)	26	Conventional adenomas	Men: Yogurt consumption ≥2 servings/week compared to no consumption	OR 0.81; 95% CI 0.71–0.94	Higher yogurt consumption was associated with lower risk of conventional adenomas but not serrated lesions among men. No association with conventional adenomas nor serrated lesions was observed among women.
								Women: Yogurt consumption ≥2 servings/week compared to no consumption	OR 0.98; 05% CI 0.88–1.09	
							Serrated lesions	Men: Yogurt consumption ≥2 servings/week compared to no consumption	OR 0.89; 95% CI 0.74–1.07	
								Women: Yogurt consumption ≥2 servings/week compared to no consumption	OR 0.92; 95% CI 0.82–1.04	
Pala et al. [27]	Adult men and women aged 35–65 years living in Italy and free from cancer at baseline (EPIC study)	Prospective cohort	14,178 (men) 31,063 (women)	Yogurt	Frequency of intake (g per day)	12	Incident CRC	Men: Highest intake tertile compared to lowest intake tertile	HR 0.47; 95% CI 0.28–0.81	Higher yogurt consumption was associated with lower relative rates of CRC. The effect was stronger among men than women.
								Women: Highest intake tertile compared to lowest intake tertile	HR 0.69; 95% CI 0.47–1.03	
Senesse et al. [28]	Adult men and women aged 30–79 years in France	Case-control	789	Yogurt	Frequency of intake (g per day)	-	Incident small adenomatous polyps	High consumption compared to no consumption	OR 1.2; 95% CI 0.8–2.1	High yogurt consumption was associated with lower relative risk of large adenomas. No association was noted between yogurt consumption and small adenomas.
							Incident large adenomatous polyps	High consumption compared to no consumption	OR 0.6; 95% CI 0.4–1.0	
Murphy et al. [29]	Adult men and women aged 35 + years participating in the EPIC study free from cancer at enrollment	Prospective cohort	477,122	Yogurt	Frequency of intake (g per day)	11	Incident CRC	Highest intake quartile compared to no intake	HR 0.90; 95% CI 0.81–0.99	Higher yogurt intake was inversely associated with risk of CRC.
Michels et al. [30]	Adult men and women aged 30–75 years participating in HPFS and NHS	Prospective cohort	43,269 (men) 83,054 (women)	Yogurt	Frequency of intake at baseline and cumulatively updated (servings per week, month)	32	Incident CRC	Baseline: 1+ serving/week compared to never or <1 serving/month	HR 0.89; 95% CI 0.80–1.0	More frequent yogurt consumption was associated with lower incidence of CRC.
								Cumulative update: 1+ serving/week compared to never or <1 serving/month	HR 0.97; 95% 0.87–1.07	

Abbreviations: EPIC: European Prospective Investigation into Cancer and Nutrition; GOS: Galacto-oligosaccharides; TCPS: Tennesee Colorectal Polyp Study; JHBS: Johns Hopkins Biofilm Study; PREDIMED: Prevencion con Dieta Mediterranea; HPFS: Health Professionals Follow-up Study; NHS: Nurses’ Health Study.

## Data Availability

Not applicable.

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
