# Peer review of "The Impact of Prebiotic, Probiotic, and Synbiotic Supplements and Yogurt Consumption on the Risk of Colorectal Neoplasia among Adults: A Systematic Review"

_nutrients, 2022, doi:10.3390/nu14224937_

Round 1
Reviewer 1 Report
The authors present a systematic population-based review over probiotic-type interventions and colorectal neoplasia. A review of this type has not yet been performed. This review covers four groups: prebiotic, probiotic, synbiotic, and yogurt supplementation in adults and if these affect colorectal neoplasia.
As use of probiotic intervention may be chemoprotective, I found this review to be relevant and of interest. It was well written and organized.
Author Response
We are very grateful for the positive review from reviewer one.
Reviewer 2 Report
The manuscript entitled "The Impact of Prebiotic, Probiotic, and Synbiotic Supplements 2 and Yogurt Consumption on the Risk of Colorectal Neoplasia 3 among Adults: A Systematic Review" by kim et al is interesting, but there are some concerns regarding this study.
The authors only quoted the results of the former studies. However, in order to improve the quality of this paper, the authors should add more studies, including possible reasons or mechanistic studies to elaborate on the reason why the yogurt is inversely associated with colorectal neoplasia. Why not the probiotics or Synbiotic supplementation? Moreover, the conclusion should cover research outcomes, application in other fields/industry/studies, and future perspective.
Round 2
Reviewer 2 Report
In the present form the paper is acceptable